# Case Report of Puffinosis in a Manx Shearwater (*Puffinus puffinus*) Suggesting Environmental Aetiology

**DOI:** 10.3390/ani12243457

**Published:** 2022-12-07

**Authors:** Niamh P. G. Esmonde, Robert E. B. Hanna, Jignasha G. Patel, Victoria J. Smyth, Paul Caplat, Wesley Smyth, Paris Jaggers, Oliver Padget, Tim Guilford, Chris Perrins, Neil Reid

**Affiliations:** 1School of Biological Sciences, Queen’s University Belfast, Belfast BT9 5DL, UK; 2Veterinary Sciences Division, Agri-Food & Biosciences Institute, Belfast BT4 3SD, UK; 3Food Safety Department, Teagasc Food Research Centre, D15 DY05 Dublin, Ireland; 4Institute of Global Food Security (IGFS), Queen’s University Belfast, Belfast BT9 5DL, UK; 5Copeland Bird Observatory, Old Lighthouse Island, Donaghadee, UK; 6Department of Zoology, University of Oxford, Oxford OX1 3SZ, UK

**Keywords:** *Clostridia perfringens*, coronavirus, disease, *Enterococcus faecalis*, *Flavobacterium*, metagenomics, necropsy, *Pseudomonas*, seabird, *Serratia*

## Abstract

**Simple Summary:**

Puffinosis is a disease of seabirds characterised by blistering and necrosis of webbed feet. It is a poorly understood, fatal disease among breeding colonies of Manx shearwaters (*Puffinus puffinus*). The causative agent has been suggested to be a coronavirus, however, the virus described in a previous study seems likely to have been a laboratory contaminant. Thus, the aetiology and mode of transmission of puffinosis remains uncertain. Using metagenomic sequencing, we found no evidence of a viral infection. Instead, common bacteria found in soil and on skin were present, suggesting an opportunistic bacterial infection is likely the cause of the blistering, perhaps entering the skin after prolonged contact with caustic faecal ammonia while sitting in a damp nesting burrow. This study demonstrates the use of metagenomic screening for infectious agents.

**Abstract:**

Puffinosis is a disease of a range of seabirds characterised by dorsal and ventral blistering of their webbed feet, conjunctivitis, dry necrosis, leg spasticity, head shaking, loss of balance, tremors, and death. It is associated with Manx shearwaters (*Puffinus puffinus*), frequently affecting chicks within their underground nesting burrows. The aetiology of the disease is unclear but has been attributed to a type-2 coronavirus associated with *Neotombicula* mites as a potential vector. However, there is some uncertainty given potential laboratory contamination with mouse hepatitis virus and failure to fulfil Koch’s postulates, with birds injected with isolates remaining healthy. We describe a detailed case report of puffinosis in a Manx Shearwater covering necropsy, histology, bacteriology, and metagenomics including viral sequencing. We found no evidence of viral infection or parasites. Our results are consistent with an entirely environmental aetiology, with caustic faecal ammonia in damp nesting burrows causing conjunctivitis and foot dermatitis breaking the skin, allowing common soil bacteria (i.e., *Flavobacterium*, *Staphylococcus* and *Serratia* spp., *Clostridia perfringens* and *Enterococcus faecalis*) to cause opportunistic infection, debilitating the bird and leading to death. A similar condition (foot pad dermatitis or FPD) has been reported in broiler chickens, attributed to caustic faeces, high humidity, and poor environmental conditions during indoor rearing, preventable by adequate ventilation and husbandry. This is consistent with puffinosis being observed in Shearwater nesting burrows situated in tall, dense, vegetation (e.g., bracken *Pteridium aquilinum*) but rarely reported in burrows situated in well-ventilated, short coastal grasslands. This proposed environmental aetiology accounts for the disease’s non-epizootic prevalence, spatial variation within colonies, and higher frequency in chicks that are restricted to nesting burrows.

## 1. Introduction

Manx shearwaters (*Puffinus puffinus*), hereafter referred to simply as ‘shearwaters’, are medium-sized pelagic seabirds of the Family Procellariidae, which circumnavigate the Atlantic Ocean breeding on uninhabited islands in north-west Europe, including those around Britain and Ireland, and overwintering along the South American, principally the Argentinian, coast [1]. Shearwaters spend most of their time at sea during migration and overwintering with limited contact with conspecifics, likely limiting disease spread and contributing to their high annual survival [2,3]. During the breeding season, shearwater colonies are susceptible to outbreaks of poxvirus [4] and chlamydia [5], but not at epizootic levels, and avian malaria (*Plasmodium* sp.), which could cause substantial mortality in many seabirds [6]. Shearwaters are presently ‘amber listed’ in the UK and a conservation priority under the EU Birds Directive 2009/147/EC. They are the designated feature of certain Special Protection Areas (SPAs), which aim to maintain breeding colonies in good conservation status that could be threatened by disease under a changing climate [7].

For decades, a disease known as ‘puffinosis’ has been reported in young shearwaters and other seabirds. Puffinosis is a well-known but poorly understood disease typically found in members of the Order Procellariiformes, however, the drivers of transmission and aetiology remain undetermined [5]. Puffinosis causes birds’ webbed feet to blister (dorsal and ventral sides), which usually progresses to necrosis, conjunctivitis, neurological symptoms including head shaking, sneezing, and convulsions, leg spasticity, and paralysis, leading to notable juvenile mortality rates among some seabird populations [8,9,10,11]. Puffinosis is known to also affect Northern Fulmars (*Fulmarus glacialis*) [12], European Storm Petrels (*Hydrobates pelagicus*), Leach’s Storm Petrels (*Oceanodroma leucorhoa*), California Brown Pelicans (*Pelecanus occidentalis californicus*), and Gentoo Penguins (*Pygoscelis papua*) [13]. The disease was first recorded in 1908 and described in a series of papers after severe outbreaks in Manx shearwaters on Skomer Island, Wales, in 1946 and 1947, with subsequent outbreaks occurring regularly as well as on nearby Skokholm Island [5,8,9,10,11]. Early puffinosis epizootics on Skomer affecting fledglings (approximately 70 days old) were only observed when they emerged from nesting burrows for the first time towards the end of the breeding season from August to mid-September. In contrast, Nuttall et al. (1982) [14] found puffinosis in young chicks and fully grown birds throughout the breeding season, though not at epizootic levels. Out of the six monitored shearwater colonies in the United Kingdom (three in Wales, two in Scotland and one in Northern Ireland), puffinosis has only been recorded on Skomer and Skokholm, Wales, and the Copeland Islands, Northern Ireland [9,11,15,16]. Despite this, there is evidence that shearwaters move between colonies [17] and cases of puffinosis have also recently been reported in France [18]. The incidence of puffinosis on Skomer and Skokholm varies with geographical location, appearing predominantly in birds within nest burrows in dense vegetation, generally bracken (*Pteridium aquilinum*), but very seldom in birds within nest burrows in open, short maritime grasslands (author O.P. pers. obs.), suggesting environmental factors may be involved [14]. 

The aetiology and mode of transmission of puffinosis in shearwaters is unclear. Dane et al. (1953) [10] suggested that puffinosis is primarily a disease of gulls transmitted to shearwaters through contact with soil or stones contaminated with blister fluid. However, the incidence of puffinosis in gulls is low [14]. Species of mites *(Trombiculata autumnalis*) and fleas (*Ornithopsylla laetitiae*) were suspected to be vectors to a possible viral agent since Stoker and Miles (1953) [19] isolated a virus from *Neotrombiculid autumnalis* similar to a virus recovered from diseased shearwaters [5,14,20]. Studies on experimental transmission and the epizootiology suggest puffinosis may be caused by a type-2 coronavirus isolated (via laboratory mouse inoculation) from blood, vesicle fluid, and *Neotombicula* mites from diseased shearwaters [21]. However, it is unclear whether this virus was a contaminant mouse hepatitis virus (MHV), which has high pathogenicity in laboratory mice [20]. Additionally, the virus found by Nuttall and Harrap (1982) [21] differed from the one found by Miles and Stoker (1948) [9] and inoculation of the virus into two shearwaters did not lead to the development of clinical signs, constituting a failure of Koch’s postulates. Therefore, the aetiology of puffinosis remains obscure.

The aim of the present study was to investigate an incidence of puffinosis in a Manx shearwater providing a detailed case report based on tissue histology, bacterial culture, and metagenomic sequencing of infected tissues, which provide clues to the potential aetiology of the disease.

## 2. Materials and Methods

### 2.1. Study Site

A fully grown Manx Shearwater (*Puffinus puffinus*) with the clinical signs of puffinosis was found on Lighthouse Island (also known as Bird Island), the Copeland Islands, Northern Ireland (54°41′42.0″ N, 5°31′29.1″ W) on 21 May 2021. This site is an Area of Special Scientific Interest (ASSI) and a Special Protection Area (SPA), originally designated under the EU Habitats and Species Directive 92/43/EEC (subsequently translated into local law since the UK left the European Union in 2020) for, among other things, its shearwater breeding colony of approximately 4900 pairs [22]. The island is managed and monitored by the Copeland Bird Observatory (www.thecbo.org.uk; accessed on 29 October 2021). It hosts a population of introduced rabbits (*Oryctolagus cuniculus*), which provide nesting burrows for shearwaters, although they are capable of digging their own [5,23]. Previously, breeding colonies were located in open, short maritime grasslands, but, in recent decades, bracken (*Pteridium aquilinum*) has expanded, replacing the grasslands with dense stands now dominating most of the island’s area, including the Shearwater colonies, where access is maintained for ringing birds by ongoing habitat management, i.e., manual and mechanical bracken cutting. Puffinosis was recorded in five birds in 1990 and one bird each in 1991 [15,16] and 2014 (unpublished data). 

The case in 2021 was a bird of unknown age, sex, and origin (it was unringed), and a diagnosis of puffinosis was made in consultation with author C.P. based on (i) blistering of the webbing of both feet (Figure 1), (ii) stereotypical head shaking movements consistent with the neurological onset of puffinosis (Appendix A), and (iii) a severe degree of paralysis (the bird was unable to fly or walk). The bird was placed in a box but died the following evening. Post-mortem, histological, parasitic, bacterial, and viral screening were conducted by the Agri-Food and Biosciences Institute (AFBI) as part of the Queen’s–AFBI Alliance. 

### 2.2. Necropsy Including Histology, Bacteriology, and Parasitology 

Necropsy was conducted to examine the skin and organs for any evidence of inflammation, structural abnormalities, wounds or lesions. At post-mortem, the following tissues were collected for histological examination: brain, liver, kidney, eye (including palpebral and bulbar conjunctiva), lesioned footpad skin, testes (the bird proved to be male upon dissection), trachea, lung, and proventriculus. All tissue samples were fixed with 10% neutral buffered formalin for 24 h prior to dehydration, embedding in wax, histological sectioning at 3 µm thickness, and staining with haematoxylin and eosin or with Grocott’s silver-impregnation method for fungal elements [24].

Blister fluid and lesioned skin from the feet were collected for electron microscopy and for metagenomic analysis. Electron microscopy was carried out on homogenised sub-samples, which had been dried on carbon-coated nickle grids and negatively stained using ammonium molybdate.

The following samples were collected for bacteriological culture: liver, lung, kidney, and lesioned skin from the foot web. All samples were cultured using blood agar (with atmospheric air, a 10% CO_2_ atmosphere, and anaerobic conditions); McConkey agar (with atmospheric air); Yersinia-selective medium (with atmospheric air); and selenite broth medium (with atmospheric air). Procedures defined by the World Organisation for Animal Health OIE regulations were followed throughout.

Small intestine contents were collected for parasitological examination for helminth eggs and coccidial oocysts using the McMaster floatation method [25].

### 2.3. Nucleic Acid Extraction, Library Preparation and Sequencing

Total RNA and DNA was isolated from a foot blister tissue sample using an RNA/DNA purification kit (Norgen Biotek Corp., Thorold, ON, Canada), following the manufacturer’s instructions. Briefly, the tissue sample was subjected to bead beating in TissueLyser II (Qiagen, Hilden, Germany) for 3 min at 30 Hz and subsequently, the homogenised sample was processed for lysis as per manufacturer′s instructions. The lysed tissue sample was centrifuged at 14,000× *g* at 4 °C for 10 min to separate tissue debris. Approximately, 200 μL of supernatant was processed from the previous step for RNA and DNA extraction separately, followed by nuclease treatment. Quantity and quality of metagenomic RNA and DNA were assessed using a Qubit 3.0 fluorometer (ThermoFisher Scientific, Loughborough, UK).

To infer pathogen species taxonomic information, as well as functional information, from the affected footpad lesion, a shotgun metagenomics library was prepared using a Sure Select XT HS2 RNA library preparation kit (Agilent, Dublin, Ireland) with minor modifications. The extracted RNA portion was processed for cDNA synthesis, which was subsequently combined with an equal proportion of fragmented DNA for library synthesis. The final library size was determined using Fragment Analyzer (Agilent) and the library concentration was measured using a Qubit fluorometer DNA HS kit (Invitrogen, Waltham, MA, USA). The sequencing run was carried out using 4 nM diluted libraries on a Miseq instrument (Illumina, San Diego, CA, USA) employing 2 × 300 bp V3 chemistry. 

Shotgun raw reads were pre-processed and assigned to the MG-RAST (Metagenomic Rapid Annotations using Subsystems Technology) server v4.0 with default parameters [26]. Singleton reads were discarded. The raw fasta file was screened for quality trimming followed by removal of artificial duplicate reads, and screening for contaminant DNA of laboratory model organisms (mice, rats, rabbits, zebra finch, etc.) and humans. The uploaded sequences were trimmed [27] with a minimum Phred score of 15 and taxonomic profiling was performed using the RefSeq database with a minimum E-value of 1E-5 and minimum identity of 70%. MG-RAST advise against the reliability of taxonomic profiling at the species level [26], in particular for samples obtained by shotgun sequencing. For this reason, we chose to perform the analysis of the microbial communities to genus level only. The analysis was not restricted to prokaryotes but also included viruses. A taxonomical hits distribution was carried out using the contigLCA algorithm [28] by finding a single consensus taxonomic entity for all features on each individual sequence.

## 3. Results

Post-mortem examination revealed the carcass to be in good condition and well feathered, but with advanced blistering and necrosis of the webbed skin of both feet with several fluid-filled blisters (Figure 1). There were signs of conjunctivitis, but the conjunctiva had no significant abnormalities. The viscera were tacky and dehydrated. The proventriculus and small intestines had dark brown/black semi-solid contents that were small in volume. No grossly apparent abnormalities were found in the brain, trachea, gizzard, heart, lungs, or kidneys. The intestinal contents revealed large numbers of coccidial oocysts. Electron microscopy failed to detect any identifiable viral particles. 

No significant histological abnormalities were detected in the trachea, liver, or lungs. The bird was male with fully formed testes and maturing sperm present. The most significant histological finding was severe vesicular necrotising dermatitis of the foot web (Figure 2) and acute palpebral conjunctivitis. Further findings included: congestion and haemorrhage in dermis; severe heterophilic dermatitis with vasculitis and thrombosis in dermal vessels; multiple bacterial colonies (medium-sized rods), mainly in superficial dermis and base of epidermis, with detachment of stratum corneum (blistering); necrosis of basal layers of epidermis in areas where stratum corneum was detached; and fungal hyphae in superficial dermis/basal epidermis. In the brain, there was a large focus of vesiculation and encephalomalacia in the cerebral white matter.

Bacteriological culture of foot blister fluid contained *Flavobacterium* and *Staphylococcus* spp. in some concentration, with *Serratia* spp. and *Clostridia perfringens* present. The liver and kidneys contained high concentrations of *Serratia* spp., while *Flavobacterium* spp. and *Enterococcus faecalis* were present in the liver only. The lungs were clear. 

Metagenomic analysis of foot blister fluid generated 33,600 sequences, totalling 8,140,019 base pairs with an average length of 242 bps. After removal of potential laboratory contamination, e.g., human and other eukaryotic reads, the majority (75.06%) were identified as Proteobacteria. The most common bacterial genus present was *Pseudomonas* (74.1% of all reads), followed by *Escherichia* (4.6%), *Acinetobacter* (2.5%), *Serratia* (1.8%), *Vibrio* (1.8%), *Listeria* (1.3%), *Shewanella* (1.2%), *Burkholderia* (1.1%), and all other genera representing <1% of all reads (Table 1). No viral (including coronavirus) nucleic acids were recovered.

## 4. Discussion

In contrast to previously suggested aetiology, this case report of puffinosis failed to return any evidence of the involvement of a coronavirus or of other viruses [21]. No ectoparasites that are potential vectors of disease, such as mites, were recovered [19]. Previous investigations into puffinosis used electron microscopy, bacterial culture, and in vivo studies to screen blood, liver, kidney, lung, and blister tissues from which viral particles have been apparently isolated [9,19,21]. However, there are concerns that the coronavirus recovered from laboratory mice inoculated with blister fluid may have been due to the mice already being host to a coronavirus, mouse hepatitis virus (MHV), providing contamination [21]. Moreover, inoculation of healthy shearwaters with blister tissue fluid [19] or recovered from lab mice [21] failed to induce clinical signs of puffinosis. This is the first study using metagenomic sequencing to screen for viruses.

In the skin of the foot, we found evidence of bacterial and fungal infection, which likely represents opportunistic colonization of previously damaged integument. Procellarids have very short tarsi and a famously poor ‘crash landing’ technique. Crashing through vegetation such as bracken, or landing on sharp stems left after bracken cutting, could cause physical trauma to the fleshy webbing of the feet. Small lacerations to the dermal tissue could provide a route for bacterial infection. Whilst the bird examined here was seemingly juvenile or adult, puffinosis is commonly reported among chicks, which have had no opportunity to physically injure their feet. Nor would this route cause other characteristic symptoms of puffinosis such as conjunctivitis. Thus, we consider it more likely that the primary lesions on the feet may be contact dermatitis caused by caustic ammonia from faecal deposition in the nesting burrow, which may also account for the conjunctivitis. *Eimeria* apicomplexan parasites (the causative agent of coccidiosis) require damp substrate conditions to sporulate and become infective [29], while all bacteria cultured from the bird are present in the environment, generally favouring damp conditions [30,31,32]. Some, including *Pseudomonas*, *Serratia*, *Staphylococcus* spp., and *Clostridium perfringens*, can act as opportunistic pathogens, especially in debilitated animals, but are unlikely to be primary pathogens (e.g., the lungs were clear of *Pseudomonas*). Faecal ammonia could cause conjunctivitis and open the skin of the sensitive webbed feet through dermatitis, allowing free-living bacteria common in soil to invade, causing opportunistic infection that spreads to the liver and kidneys. Inability to fly prevented normal feeding activity, which may have ultimately led to dehydration and starvation (as observed in infections in Great Tits (*Parus major*) [33] or American Flamingos (*Phoenicopterus ruber*) [34]). The lesion seen in the brain could underlie neurological symptoms such as head shaking, sneezing, or convulsing and may have been related to debility and circulatory failure. Dane (1948) [8] noted that it was common to find shearwaters showing signs of puffinosis with spastic leg extensions. Brain lesions in birds have been observed to disrupt calling behaviour [35] and impair mobility [36]. An environmental aetiology is consistent with current and previous infections elsewhere associated with nesting burrows in dense, dank vegetation, i.e., bracken cover, but less frequent in burrows in open, short grasslands [14]. Atlantic Puffins (*Fratercula arctica*) have a similar breeding biology to shearwaters [37,38] yet puffinosis has not been reported in puffins, perhaps because they avoid nesting near dense vegetation as predators, such as Lesser Black-backed Gulls (*Larus fuscus*), Great Skuas (*Stercorarius skua*) and Great Black-backed Gulls (*Larus marinus*), may use it as cover to attack from. Burrows in short grasslands are more likely to remain drier with better air circulation and, thus, less likely to facilitate environmental conditions conducive to damp, dirty conditions. Puffins nest close to cliff edges and slopes, which typically have short vegetation where burrows are well ventilated.

A strikingly similar illness to puffinosis has been reported in broiler chickens: foot pad dermatitis (FPD), a condition causing necrotic lesions on the plantar surface of the foot, often leading to leg and foot abnormalities, and a decrease in carcass quality [39,40]. FPD clinical signs have been attributed to poor environmental conditions in broiler houses where atmospheric and ground moisture levels, due to respiration and excretion, exacerbate the impact of faecal ammonia burns. Prolonged contact with caustic faecal ammonia can cause severe dermatitis to the feet and is considered painful for birds, consequently restricting their mobility [41]. Lesions caused by FPD are associated with bacterial infections through opportunistic microbial activity, such as *Staphylococcus aureus* and *Escherichia coli*, which are present in chicken house litter and on birds’ skin [42]. Adequate broiler house ventilation and clean litter reduce the incidence and severity of ammonia burns [43,44]. In comparison, the nesting burrows for shearwaters are often among tall and dense vegetation, limiting ventilation, and in addition, non-fledged shearwaters have little opportunity to clean their feet and are unable to wash in saltwater, with the disease most commonly reported among chicks. Puffinosis might, therefore, be expected to be present during the incubation period from May to June, lasting on average 51 days [37], when adults spend more time in burrows and less frequent during the chick-rearing stage when adults spend more time at sea. In this case report, the fully grown bird was recovered from the colony surface so we cannot be certain the bird acquired the infection whilst in a burrow. We found no evidence that the presentation of puffinosis was associated with a virus or vector-borne pathogen. Therefore, we propose the hypothesis that puffinosis in Manx shearwaters might be an environmental condition most common in nesting burrows surrounded by tall dense vegetation and, thus, more likely to have damp internal conditions than those in more open, well-ventilated habitats. Faecal build-up causes dermatitis to the delicate webbed feet, breaking the skin and allowing soil bacteria to cause opportunistic infections, debilitating the bird, which, if juvenile or adult, may suffer from a decline in body condition due to reduced foraging and dehydration. This accounts for non-epizootic prevalence, spatial variation within colonies, association with nesting burrows within bracken, and higher frequency in chicks, which are restricted to the nesting burrow. Further studies of puffinosis should focus on faecal deposits in burrows, soil moisture, and burrow humidity, related to burrow position and aspect with respect to vegetation, in order to test the aetiology suggested here. Further use of metagenomic sequencing would enhance disease screening.

## 5. Conclusions

The use of metagenomics in this study helped to inform the search for the aetiology of puffinosis. Our study suggests that puffinosis may be associated with the environmental conditions shearwaters experience when in the burrow for a prolonged period during the breeding season, and, in this case, possibly egg incubation. The lack of vector and viral presence suggests the disease is unlikely to be linked to a virus or vector-borne pathogen. Infection seems likely due to opportunistic bacteria already present in the environment causing blisters and neurological symptoms after the skin of the webbed feet are burned from prolonged contact with ammonia from accumulated excreted waste. Understanding such causes of this disease maybe important for assessing and providing effective habitat management strategies to prevent outbreaks, i.e., removal of rank invasive bracken and the ecological restoration of short maritime grasslands to increase burrow ventilation. The association between puffinosis in shearwaters, faecal deposits in the burrow, and dense vegetation needs further examination.

## Figures and Tables

**Figure 1 animals-12-03457-f001:**
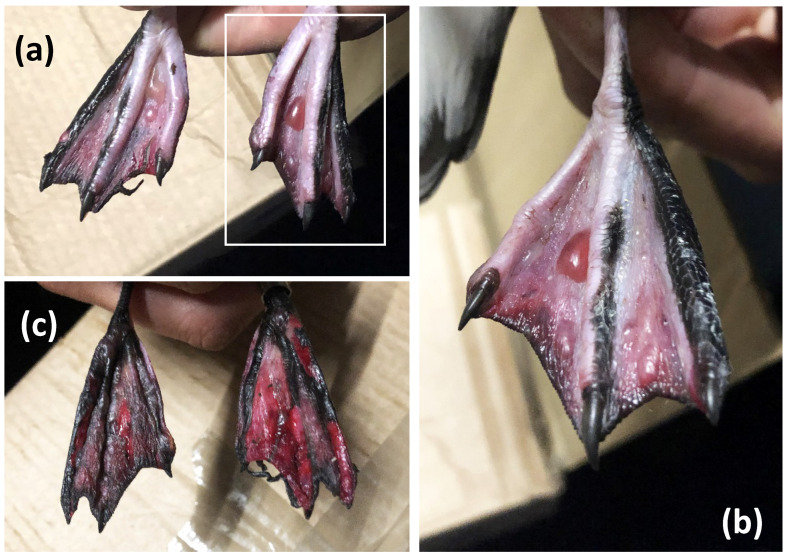
Blistering and necrosis of the foot web of a Manx Shearwater (*Puffinus puffinus*). (**a**) Dorsal surface of both feet showing epidermal blistering. (**b**) A close up of the blistering on the left foot as shown in the insert. (**c**) The ventral surface of both feet showing necrosis. Images © Paris Jaggers.

**Figure 2 animals-12-03457-f002:**
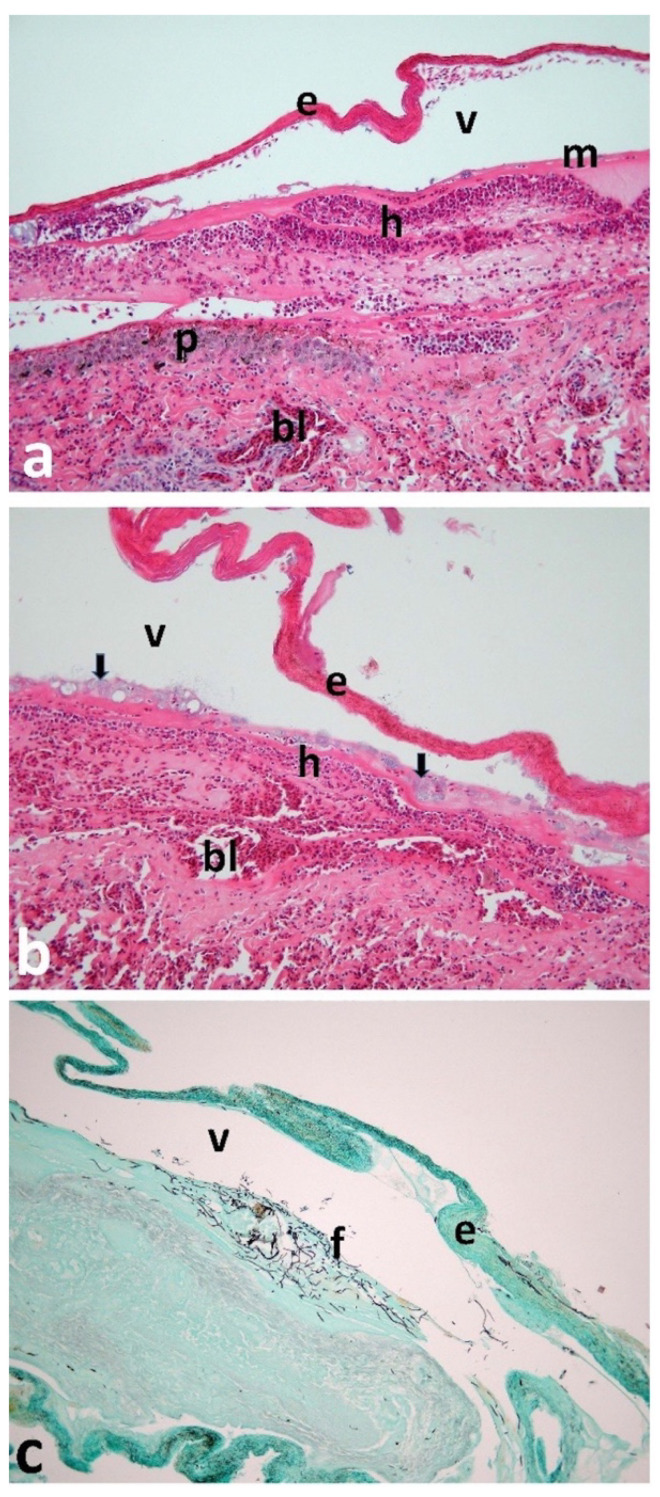
Light microscope images (×100 magnification using a Leica Pathology microscope fitted with a Nikon Coolpix camera) of sections of the foot web skin. Sections (**a**,**b**) were stained with haematoxylin and eosin (H&E) for general histology, while section (**c**) was stained with Grocott’s silver-impregnation technique to display fungal elements. (**a**) The keratinised layer of the epidermis (e) was elevated from the basal lamina (m) by a fluid-filled vacuole (v), forming a blister. In the superficial dermis, many heterophils (h) were massed together, representing an acute inflammatory reaction. Pigment-containing cells (p) were also present in the dermis, and numerous small dermal blood capillaries (bl) were congested with erythrocytes. (**b**) Numerous blue-grey-stained bacterial colonies (arrows) were present at the base of the epidermis and in the superficial dermis. The keratinised layer of the epidermis (e) was elevated by the blister fluid (v), and congested blood vessels (bl) were present throughout the dermis. Heterophilic inflammation (h) was visible in the superficial dermis. (**c**) Branching fungal mycelium (f) was present in the superficial dermis, beneath the blister formed by the epidermal flap (e) and the vesicle fluid (v). Images © Robert Hanna.

**Table 1 animals-12-03457-t001:** Metagenomic sequence taxonomic hit distribution accounting for 95% of all reads (cumulative percentage abundance) showing alignment length and % identity (certainty in identification). Note only bacteria were identified (74% *Pseudomonas*) with no viral nucleic acids present.

Phylum	Class	Order	Family	Genus	Alignment Length	% Identity	Count Abundance	% Abundance	Cumulative % Abundance
Proteobacteria	Gammaproteobacteria	Pseudomonadales	Pseudomonadaceae	*Pseudomonas*	68	90.9	563	74.1	74.1
Proteobacteria	Gammaproteobacteria	Enterobacteriales	Enterobacteriaceae	*Escherichia*	91	87.6	35	4.6	78.7
Proteobacteria	Gammaproteobacteria	Pseudomonadales	Moraxellaceae	*Acinetobacter*	75	82.5	19	2.5	81.2
Proteobacteria	Gammaproteobacteria	Enterobacteriales	Enterobacteriaceae	*Serratia*	53	94.7	14	1.8	83.0
Proteobacteria	Gammaproteobacteria	Vibrionales	Vibrionaceae	*Vibrio*	36	74.1	14	1.8	84.9
Firmicutes	Bacilli	Bacillales	Listeriaceae	*Listeria*	29	75.9	10	1.3	86.2
Proteobacteria	Gammaproteobacteria	Alteromonadales	Shewanellaceae	*Shewanella*	30	87.7	9	1.2	87.4
Proteobacteria	Betaproteobacteria	Burkholderiales	Burkholderia	*Burkholderia*	61	79.1	8	1.1	88.4
Firmicutes	Bacilli	Lactobacillales	Enterococcaceae	*Enterococcus*	64	96.2	7	0.9	89.3
Proteobacteria	Gammaproteobacteria	Aeromonadales	Aeromonadaceae	*Aeromonas*	55	97.3	5	0.7	90.0
Proteobacteria	Betaproteobacteria	Burkholderiales	Oxalobacteraceae	*Janthinobacterium*	66	87.9	5	0.7	90.7
Firmicutes	Clostridia	Clostridiales	Clostridiaceae	*Clostridium*	63	74.6	4	0.5	91.2
Proteobacteria	Alphaproteobacteria	Rhizobiales	Beijerinckiaceae	*Beijerinckia*	38	70.5	3	0.4	91.6
Proteobacteria	Gammaproteobacteria	Unclassified	Unclassified	*Endoriftia*	54	71.9	3	0.4	92.0
Proteobacteria	Betaproteobacteria	Burkholderiales	Oxalobacteraceae	*Herminiimonas*	57	84.8	3	0.4	92.4
Proteobacteria	Gammaproteobacteria	Enterobacteriales	Enterobacteriaceae	*Pantoea*	66	91.4	3	0.4	92.8
Proteobacteria	Gammaproteobacteria	Enterobacteriales	Enterobacteriaceae	*Providencia*	25	92.0	3	0.4	93.2
Proteobacteria	Gammaproteobacteria	Enterobacteriales	Enterobacteriaceae	*Yersinia*	61	85.1	3	0.4	93.6
Proteobacteria	Alphaproteobacteria	Rhizobiales	Rhizobiaceae	*Agrobacterium*	98	79.0	2	0.3	93.8
Proteobacteria	Gammaproteobacteria	Pseudomonadales	Pseudomonadaceae	*Azotobacter*	45	77.2	2	0.3	94.1
Proteobacteria	Betaproteobacteria	Burkholderiales	Alcaligenaceae	*Bordetella*	62	90.3	2	0.3	94.3
Proteobacteria	Betaproteobacteria	Burkholderiales	Burkholderiaceae	*Cupriavidus*	66	89.4	2	0.3	94.6
Proteobacteria	Gammaproteobacteria	Enterobacteriales	Enterobacteriaceae	*Enterobacter*	53	72.0	2	0.3	94.9
Spirochaetes	Spriochaetes	Spirochaetales	Leptospiraceae	*Leptospira*	77	74.0	2	0.3	95.1

## Data Availability

Not applicable.

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
