# Peer review of "Case Report of Puffinosis in a Manx Shearwater (Puffinus puffinus) Suggesting Environmental Aetiology"

_animals, 2022, doi:10.3390/ani12243457_

Round 1

Reviewer 1 Report

Authors present a novel perspective and hypothesis about a disease aetiology (puffinosis) in a Manx shearwater. I believe the case report is well-structured and provides an integrated analysis of all the findings they obtained from the necropsy, questioning what we know so far about this disease. The case is well-discussed as well and the authors sustained their perspectives and interpretations in a clear way, so it is the kind of case report that should be read.

Therefore, I only have some suggestions for the authors that I believe would improve their manuscript:

1) From my personal perspective the introduction is a little bit too long. It is not necessary to debate diseases of seabirds at first. It is not wrong, of course, but for a case report I would introduce the host species (other common diseases reported in thar species as well), the pathogen or disease you are studying and the importance of studying it (conservation?, One Health?, captivity management?, welfare?). Moreover, this is not only applied to seabirds diseases, and since it appears after the presentation of other seabird diseases it seems to be only related to them: "However, the means by which a disease can cause an outbreak or emerge are also dependent on the mechanism and drivers of disease transmission, which involves several factors: host susceptibility, environmental and anthropogenic factors, population density, pathogenicity and virulence, and co-infections."

2) I believe the last paragraph of the introduction should clearly present the aim of the study. If I understood correctly, you intend to present us this case report and use it to discuss the true aetiology of puffinosis, confronting what is commonly assumed. This idea is there but I would personally rephrase it a little bit.

3) L (308-310) "Eimeria apicomplexan parasites (the causative agent of coccidiosis) require damp substrate conditions to sporulate and become infective while all bacteria cultured from the bird are present in the environment, generally favouring damp conditions" - This sentence requires a reference.

4) I am not an expert in virology analysis. But after reading your manuscript one of the questions I have in my mind is if there is anything in your sampling method that can explain why you didn't find a virus. Other authors found very different viruses and vectors associated with similar clinical conditions, so I totally agree that the transmission of the disease is not clear and environmental aetiology is probably causative. But could have been any mistakes in your methods that justify not finding viruses at all? Your sampling method (foot blister fluid) is the same used by other authors? I would clarify this in M&M or discuss it in the Discussion section, it would give strength to your report.

I have nothing  further to comment on this manuscript.

Author Response

Please see the attachment. Many thanks.

Reviewer 2 Report

This paper is generally well written, but will benefit from a few changes to wring style.  When listing multiple factors it is more conventional to simply use the conjunction 'and' between the last two factors, rather than '... , and ..."

The titles of the references cited are nearly all capitalized, which may not meet the journals style guide requirements, but it creates the added problem of altering latin binomials and trinomials to an incorrect format,  For example see the citations mentioning Puffinus p. puffinus become Puffinius p. Puffinus.

Also all species names  provided in latin should be in italics.

Specific suggestions are as follows:

L18 Delete the first comma after 'sequencing'

L20 Delete the first use of the word 'after' as it is not required.

L34 Add 'and' after 'bird'.

L48 Change 'from' to 'by',

L56 Change 'to' to 'for'.

L57 Delete the comman after '[6].

L62 Change '(Puffinus puffinus) to '(P. puffinus)' as it is a subsquent use of the binomial.

L106 Add '[22]' after '(1982)'.

L110 Add '[17]' after '(1953)'.

L114 Add '[36]' after Slater and Miles (1953)' renumber to '[23]' and then (?) renumber all new citations after this and check the sequence/ or change the order of citations in the references.

L120 Add '[28]' after '(1982)'

L121 and L 321 Add '[16]' after '(1948)'.

L308 'apicomplexan' should be in plain text and not italics - it isn't a species name.

L349 Add 'are' after 'and'.

I think the association of reported cases of this disease with nests located in habitat supporting bracken fern could also have a few more interpretations.  First, bracken fern can grow in soils that have a wide range in pH (from 2.8-8.6).  Acidic soils could interact with bird faeces and contribute to/or facilitating the blistering of the skin of the birds' feet.  The higher humidity in bracken patches would help facilitate any chemical reaction between the faeces and low pH soils when present. 

Second, Procellarids have vey short tarsi and a famous for their poor landing technique.  If the shearwaters are 'crash landing' into the bracken patches before making their way to their nest burrows on foot they are likely experiencing physical trauma to the fleshy webbing of their feet.  Small lacerations to the dermal tissue would then provide a ready route for bacterial infection -with or without the increased humidity or low soil pH. The cut or broken stems of managed bracken would exacerbate this process as these cut stems do not break down readily unless mechanically mulched, and this would also explain why there are more lesions on the ventral surface of the feet., and they seem to be concentrated in the areas of webbing between the toes rather than only the line of the toes, as might be expected given the toes bear most of the bird's weight when it is walking.  If this is the case then there are implications for how the stands of bracken are managed and alternatives to physical manipulation might need to be considered.

Author Response

Please see the attachment. Many thanks.
